# An Empirical Study on Disentanglement of Negative-free Contrastive Learning

**Jinkun Cao**[3]    **Ruiqian Nai**[1]    **Qing Yang**[4]    **Jialei Huang**[1]    **Yang Gao**[1,2 *]

Tsinghua University[1]    Shanghai Qi-Zhi Institute[2]
Carnegie Mellon University[3]    Shanghai Jiao Tong University[4]

## Abstract

Negative-free contrastive learning methods have attracted a lot of attention with simplicity and impressive performances for large-scale pretraining. However, its disentanglement property remains unexplored. In this paper, we examine negative-free contrastive learning methods to study the disentanglement property empirically. We find that existing disentanglement metrics fail to make meaningful measurements for high-dimensional representation models, so we propose a new disentanglement metric based on Mutual Information between latent representations and data factors. With this proposed metric, we benchmark the disentanglement property of negative-free contrastive learning on both popular synthetic datasets and a real-world dataset CelebA. Our study shows that the investigated methods can learn a well-disentangled subset of representation. As far as we know, we are the first to extend the study of disentangled representation learning to high-dimensional representation space and introduce negative-free contrastive learning methods into this area. The source code of this paper is available at `https://github.com/noahcao/disentanglement_lib_med`.

## 1 Introduction

Existing disentangled representation learning methods are mostly generative models (25; 15; 24; 10; 8; 31). They are evaluated only on simple synthetic datasets (36; 5) and low representation dimensions, e.g., no more than 20-d. In contrast, contrastive learning is a class of discriminative methods, trained by pulling the representation of two augmentations of the same image close. This usually requires a much higher representation dimension for training, e.g., at least 1000-d.

Despite the success of contrastive learning, it remains unknown whether it can lead to disentangled representations. Recently, some works (50; 53) reveal that contrastive learning can approximately invert the data generation process and allow the learned representation to have identifiability, which is related to the disentanglement property. However, all these advances necessarily rely on the contrast provided by negative samples. The disentanglement property of contrastive learning without negatives, or "non-contrastive self-supervised learning", remains unexplored. And no contrastive learning methods have been evaluated on the standard disentanglement benchmarks (36; 5) yet.

Representation disentanglement requires each single dimension of a representation vector to be correlated to only one independent variable of the input data, which we call a data factor. There lacks a uniform convention of measuring "correlation" here, so multiple disentanglement metrics (9; 15; 13) have been proposed for the quantitative evaluation. Their shared principle is that a well-disentangled representation model should not have a representation dimension responding to the change of more than one data factor. We find although multiple disentanglement metrics' (15; 24; 6; 9; 27) evaluations are in agreement for low-dimensional space (34), they disagree in high-dimensional space. Moreover, their design limitation makes the measurement not meaningful in high-dimensional space. We thus propose a new disentanglement evaluation metric that is robust to representation dimension scaling

---

*corresponding author

36th Conference on Neural Information Processing Systems (NeurIPS 2022).

and is thus adaptable to a high-dimensional representation model. The new metric is named as "**M**utual information based **E**ntropy **D**isentanglement" and MED for short.

With the proposed metric, we find that negative-free contrastive learning methods can achieve good disentanglement in a subset of latent dimensions. Given that the recent disentanglement studies are limited to simple synthetic datasets but contrastive learning is especially powerful for complicated and large-scale datasets, we also extend the quantitative benchmarking to a real-world dataset CelebA (33). On CelebA, existing low-dimensional generative disentangled learning methods are unable to learn good representations, demonstrating the gap between current disentangled learning research and real-world data complexity. To summarize, our contributions in the work are three-fold:

1. We find that existing disentanglement metrics fail to extend to high-dimensional representation space, and we propose a new metric MED to extend this area to high-dimensional space.

2. We extend the study of disentangled representation learning to real-world complicated datasets and high-dimensional representation space, revealing the gap between current disentangled representation learning and real-world data complexity.

3. We empirically study the disentanglement property of contrastive learning without negatives. We find it can learn a well-disentangled subspace of latent representation.

## 2 Related Works

**Disentangled Representation Learning.** Learning a disentangled representation is a long-desired goal in the deep learning community (4; 37; 11; 3; 42; 28; 47). A disentangled representation matches how humans understand the world and allows us to use much fewer labels to solve challenging downstream tasks (49). There are two lines of work related to this goal, i.e., the Independent Component Analysis (ICA) and the Disentangled Deep Representation Learning. ICA (18) usually assumes that the pattern of noise (17; 22) or some additional auxiliary variables (19; 23) can be observed. On the other hand, deep representation learning makes no explicit assumption on the noise distribution or representation prior and usually emphasizes unsupervised learning. Under this setting, deep representation learning is usually based on deep generative models such as VAE-based methods (15; 24; 6; 27) and Generative Adversarial Networks (GAN) (10; 8). These two lines of study focus on different but relevant (26) aspects of a learned representation. In this paper, we focus on empirically studying the disentanglement property of negative-free contrastive learning methods to introduce them into the scope of disentangled representation learning.

**Disentanglement Metrics.** The variation in the metric used also shows the difference between ICA and Disentangled Deep Representation Learning. ICA aims to achieve good identifiability and uses Mean Correlation Coefficient (MCC) as the common metric. On the other hand, the metrics used in the deep disentangled representation learning community are very diverse. Until now, no widely accepted definition of "disentanglement" is available. So the empirical agreement of metrics makes the basis of quantitatively evaluating disentanglement. DisLib (34) summarizes six popular disentanglement metrics, i.e., DCI (9), SAP (27), MIG (6), BetaVAE score (15), FactorVAE score (24) and Modularity (41). DisLib finds that the five metrics except for Modularity have good agreement in evaluating disentanglement quality. However, all of the metrics are evaluated on low-dimensional latent space, which is around 10 dimensions. We find severe problems when applying those metrics to high-dimensional latent spaces where they show significant disagreement. To make a meaningful evaluation of the disentanglement property of contrastive learning methods, we thus propose a new metric, which is more applicable to high-dimensional representation space.

**Contrastive Learning and Representation Disentanglement.** Contrastive learning (CL) creates "views" by augmentations over images. Views of the same image serve as positives, and views of other images as negatives. Recently, some works try to understand contrastive learning theoretically (50; 1; 46; 45; 30) or empirically (43; 52; 38). Zimmermann et al. suggests that the contrastive method inverts a data generation process when infinite negative samples are available (50), which is related to the disentanglement property. However, there is still no work connecting CL methods with standard disentanglement benchmarks. On the other hand, negative-free CL methods (40; 7; 51) do not use the contrast between positive and negative samples to encourage discriminative representations. They generate positive "views" of data by applying different augmentations to the same input data, and the pair of views is forwarded into two network streams. Recent works on this line have their own unique designs. BYOL (40) discovers that self-supervised learning can avoid trivial solutions, i.e.,

"model collapse", even without using negative samples to provide contrast. The key of BYOL is to add a predictor layer following the commonly adopted "encoder-projector" network of contrastive learning methods (14). This provides additional asymmetry. As a follow-up, SimSiam (7) removes the momentum update from BYOL and proves that combining the predictor and the stop-gradient design is enough for self-supervised learning to learn non-trivial representation. More recently, Barlow Twins (51) shows that even the predictor or the trick of stop-gradient is not necessary. Negative-free CL methods have their own advantages, such as avoiding model collapse (44; 16; 21), but its disentanglement property remains unexplored, either empirically or theoretically. In this paper, our focus thus is to benchmark the disentanglement property of these methods for the first time which is possible after we propose a new disentanglement metric applicable to the high-dimensional representation models.

## 3 The Proposed Disentanglement Metric: MED

Positive-negative contrast in self-supervised learning is seen to encourage learned representation to have uniformity on a hyper-sphere (50). When negative views are not available, the disentanglement property of the learned representation remains a mystery. Therefore, we examine the mentioned negative-free contrastive learning methods in an empirical study to reveal this property of interest. However, we demonstrate in Section 3.1 that existing disentanglement metrics can not evaluate CL methods in high-dimensional space fairly or meaningfully. Therefore, we introduce our proposed MI-based Entropy Disentanglement score (MED) in Section 3.2 and its variant to evaluate the disentanglement of a subspace of representation in Section 3.3.

### 3.1 Failure of Existing Metrics on High-Dimensional Space

Typical contrastive learning methods need a high dimensional representation space ("latent space") to train well. However, the previous study of disentangled representation learning only deals with low-dimensional representation space. For example, in Locatello et al., the latent space dimension is set to 10. So, existing disentanglement metrics (see Appendix C for details) are designed for low-dimensional representation models and have intrinsic flaws when evaluating models in high-dimensional space. To be precise, we have observations below:

- Metrics based on learnable classifiers, such as **BetaVAE score** and **FactorVAE score**, allow unfair advantages to a high-dimensional model whose redundant parameters can trick the classifier more easily. For example, a randomly initialized 1000-d model could reach a FactorVAE score of 61.4 on dSprites, close to many well-trained 10-d VAE-based models' scores (see Table 4 in Appendix).

- Metrics taking only one or two dimensions into score calculation, such as **SAP** and **MIG**, are biased to representations of different dimensions. Because a higher dimension makes it harder for an informative dimension to stand out and enjoy a large informativeness gap over other dimensions.

- **DCI Disentanglement score** uses a learnable regressor to score the importance of each latent dimension to each data factor. The learnable regressor, such as Gradient Boosting Tree (GBT), encourages sparsity in the output importance matrix (see Figure 7 in Appendix). So, it also gives an advantage to high-dimensional models, making it unfair to compare models of different latent dimensions. Moreover, the construction of regressors is time-intensive in high-dimension space. For example, it usually takes hours to evaluate a 1000-d representation model by DCI using GBT.

These flaws are demonstrated by our experiments, showing that existing metrics disagree for high-dimensional representations. Besides our conceptual justification of the failure of these existing disentanglement metrics, we also construct some scenarios where their failure is theoretically demonstrated in Appendix F. Overall, through our experimental evidence and theoretical justification, we show that existing disentanglement metrics can no longer make meaningful disentanglement measurements in the high-dimensional representation space.

### 3.2 Mutual Information based Entropy Disentanglement

Given the bias and limitations of existing disentanglement metrics and the necessity of high-dimensional representations for contrastive learning, we propose a new disentanglement metric for high dimensional latent spaces, which we name as "**M**utual Information based **E**ntropy **D**isentanglement", or MED in short. The calculation of MED is based on mutual information (MI) between latent dimensions and the set of data factors of input samples. MI is a widely accepted tool to measure the correlation of variables and is not biased to the models of different dimensions. Given

a dataset generated with $K$ ground truth factors $\boldsymbol{v} \in \mathbb{R}^K$ and a representation vector $\boldsymbol{c} \in \mathbb{R}^D$ , we construct an importance matrix $R \in \mathbb{R}^{D \times K}$ defined by

$$R_{ij} = I(\boldsymbol{c}_i, \boldsymbol{v}_j) / \sum_{d=0}^{D-1} I(\boldsymbol{c}_d, \boldsymbol{v}_j), \tag{1}$$

where $I(\boldsymbol{c}_i, \boldsymbol{v}_j)$ denotes the mutual information between the $i^{th}$ latent dimension $\boldsymbol{c}_i$ and the $j^{th}$ ground truth factor $\boldsymbol{v}_j$. Here, each row denotes a representation dimension and each column represents a ground truth factor. We normalize the mutual information by columns, such that an entry in the matrix indicates the relative importance of one dimension over all dimensions regarding a certain data factor. This normalization is necessary since different dimensions may have different overall informativeness to all factors. This operation fixes the gap between models of different dimensions by estimating the relative importance of a single dimension.

After normalizing over the columns, we evaluate the contribution of a dimension to different factors, which is described by a row of $R$. If one dimension is informative to only one ground truth factor, then this dimension is perfectly disentangled. This matches the mechanism of entropy. So we use the entropy to describe the disentanglement level of a dimension. We treat each row as a discrete distribution over factor index by normalization: $P_{ij} = R_{ij} / \sum_{k=0}^{K-1} R_{ik}$, where higher probability $P_{ij}$ indicates that the dimension $\boldsymbol{c}_i$ encodes more information of the factor $\boldsymbol{v}_j$. Then the disentanglement score $S_i$ for a latent dimension $\boldsymbol{c}_i$ is calculated as

$$S_i = 1 - H_K\left(P_{i\cdot}\right), \tag{2}$$

where $H_K\left(P_{i\cdot}\right) = -\sum_{k=0}^{K-1} P_{ik} \log P_{ik}$ is the entropy. $S_i$ will be higher if $\boldsymbol{c}_i$ exhibits more informativeness to one factor while less relevance to other factors. Finally, to summarize the overall disentanglement of a representation model, MED score is the weighted average of the disentanglement scores for all dimensions as

$$\text{MED}\left(\boldsymbol{c}\right) = \sum_{i=0}^{D-1} \rho_i S_i \tag{3}$$

where $\rho_i = \sum_j R_{ij} / \sum_{ij} R_{ij}$ is the relative importance of each dimension.

Our proposed MED does not use a learnable classifier or a regressor. Further, it inherits a DCI-style normalized importance matrix by taking all dimensions into account instead of using only one or two dimensions. These characteristics allow MED to be more robust to the latent dimensionality and to be better suited to meaningful disentanglement measurement in high-dimensional space. Besides the advantages MED has, it is also very computationally efficient: for a 1000-d BYOL representation model on Cars3D, the evaluation with DCI with Gradient Boost Tree by DisLib takes more than 14 hours, while MED only takes less than 20 seconds on the same machine.

### 3.3 Partial Disentanglement Evaluation Metric

It is challenging to learn a fully disentangled representation by high-dimensional models without explicitly encouraging disentanglement, especially when there are fewer independent data factors than the number of latent dimensions. However, if a high-dimensional model has a subset of dimensions that disentangle well, it is still worth studying. Such a subset can serve as a proxy to build a more compact representation model when it is hard to train such a model directly. This motivates us to design a version of MED to evaluate the partial disentanglement, which we name "Top-k MED".

The main difference is that we pick the most disentangled $k$ dimensions for each factor and compute MED on this subset of representations. For each ground truth factor $\boldsymbol{v}_j$, $\mathcal{G}_j = \{i | \arg\max_m R_{im} = j\}$ is the set of latent dimensions emphasizing this factor. Then we pick the top $k$ latent dimensions with the highest disentanglement scores in each $\mathcal{G}_j$ to construct a subset $\mathcal{P}_j = \{i | S_i \geq S^k, \ i \in \mathcal{G}_j\}$. Here, $S_i$ is the disentanglement score of the latent dimension $\boldsymbol{c}_i$ defined in Equation 2. $S^k$ is the $k^{th}$ highest disentanglement score in $\mathcal{G}_j$. Finally we obtain the subset $\mathcal{P} = \cup_j \mathcal{P}_j$. We take the sub-vector after selection $\tilde{\boldsymbol{c}} = \{\boldsymbol{c}_i\}_{i \in \mathcal{P}}$ as the representation to evaluate the MEDscore. The top-k MED is defined as MED($\tilde{\boldsymbol{c}}$).

## 4 Understanding the learned representation

In this part, we qualitatively study the disentanglement of the learned representation by negative-free contrastive learning with BYOL as an example. Since contrastive methods are not generative models, it is hard to directly do factor-controlled pixel-wise reconstruction for visualization. Instead, we

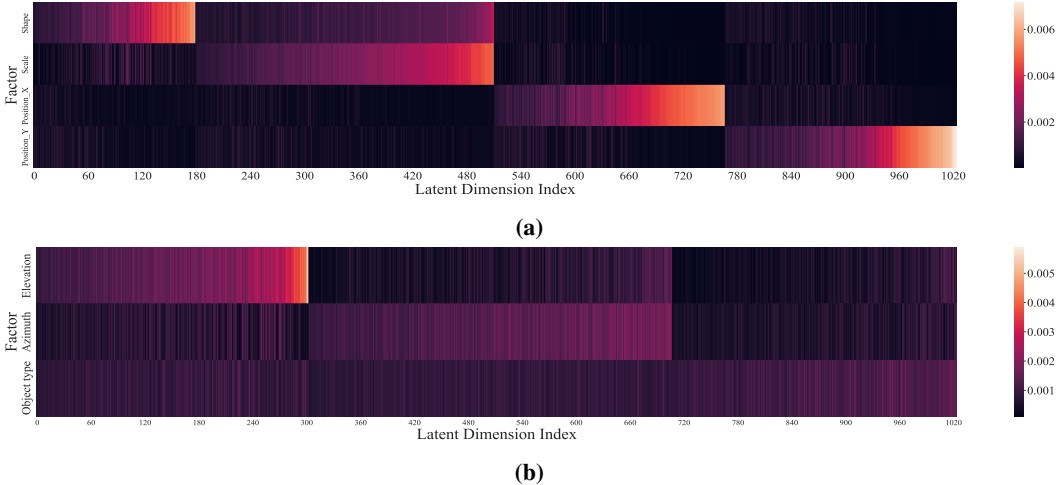

**(a)**

**(b)**

**Figure 1:** The mutual information heatmap between factors and the BYOL's latent dimensions on dSprites (a) and Cars3D (b). Each row is a ground truth factor and each column is a learned latent dimension. We normalize each row as we do in MED. Brighter straps denote higher MI. Latent dimensions are sorted for visualization.

measure the mutual information between the learned representations and ground truth factors, which is also the foundation of our proposed MED metric. More qualitative study is available in Appendix B.

### 4.1 Correspondence of Representation and Factor

To understand the disentanglement of a representation model, a basic question is how the representation dimensions correspond to data factors. After encoding an input image to a representation vector, we compute the normalized mutual information (MI) between each ground truth factor and each representation dimension, i.e. $R$ in MED defined as in Equation 1, to measure the correspondence.

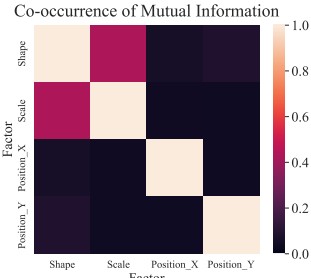

**Figure 2:** The visualization of normalized co-occurrence of mutual information on dSprites.

The mutual information between dimensions and the factors is included in Figure 1. The heatmaps are actually the transpose of the importance matrix $R^\top$. As shown by the brightness of the entries, the informativeness of the latent dimensions varies greatly. In fact, we can identify three types of columns: columns with *single*, *multiple*, and *no* bright elements. They correspond to three types of latent dimensions: *disentangled* dimensions, *entangled* dimensions, and *uninformative* dimensions. The *disentangled* dimensions only capture information about one factor while the *entangled* dimensions encode multiple factors. In contrast, the *uninformative* dimensions cannot represent factors independently. As there are still *entangled* and *uninformative* dimensions, the representations are not fully disentangled. However, by extracting the subset of *disentangled* dimensions we can derive a well-disentangled subspace.

First, we analyze the model on dSprites dataset (36). dSprites has five factors (shape, scale, orientation, position_x, and position_y). But the orientation is ill-defined with ambiguity. For example, it is impossible to distinguish if a square rotates 0 degrees or 180 degrees. As shown in Figure 1a, the *disentangled* dimensions occupy a significant proportion, indicating an evident partially disentangled pattern. We note that the degree of disentanglement varies on different datasets. An example on Cars3D is shown in Figure 1b. Cars3D is a dataset with 183 different car objects rendered from 4 elevations and 24 azimuths and the ground truth factors are not fully independent of Cars3D (see Appendix B). It is extremely hard to represent its azimuth and the type of cars with few dimensions. Thus its object-type row and azimuth row in Figure 1b are more spread out among multiple latent dimensions. This also shows the difficulty of understanding the disentanglement of the high-dimensional representation model on complicated datasets. Therefore, we will continue to conduct a quantitative evaluation with our proposed MED metric in Section 5. Further quantitative studies on more datasets are available in Appendix B.

## 4.2 Uniqueness of Factor-Representation Correspondence

In the ideal pattern of disentanglement, a representation dimension should uniquely correspond to only one factor. Now we show to what extent multiple factors are responded to by a single representation dimension in a well-disentangled subspace. Given the mutual information between the representation and the $i^{th}$ factor, noted as $I_i$, i.e., the $i^{th}$ row in the top-k version of Figure 1a, a good indicator of the uniqueness of factor-representation correspondence is the normalized co-occurrence of mutual information between the $i_1^{th}$ factor and the $i_2^{th}$ factor, which is defined as

$$\widehat{C}_{i_1,i_2} = \frac{\langle I_{i_1}, I_{i_2} \rangle}{||I_{i_1}||_2 \cdot ||I_{i_2}||_2} = \frac{\sum_{d=0}^{K \cdot k - 1} I(\tilde{\boldsymbol{c}}_d, \boldsymbol{v}_{i_1}) I(\tilde{\boldsymbol{c}}_d, \boldsymbol{v}_{i_2})}{||I_{i_1}||_2 \cdot ||I_{i_2}||_2}, \tag{4}$$

where $\tilde{c}$ is the representation after the selection process in Section 3.3. We visualize the normalized co-occurrence of mutual information among the four factors by the learned representation in Figure 2. It agrees that a dimension usually encodes only one factor. Moreover, it indicates that the learned representation tends to encode shape and scale together, which also agrees with the intuitive analysis of the independence of factor pairs. For example, the shape and scale of dSprites objects are not disentangled and independent because objects with the same scale value but in different shapes have different pixel areas.

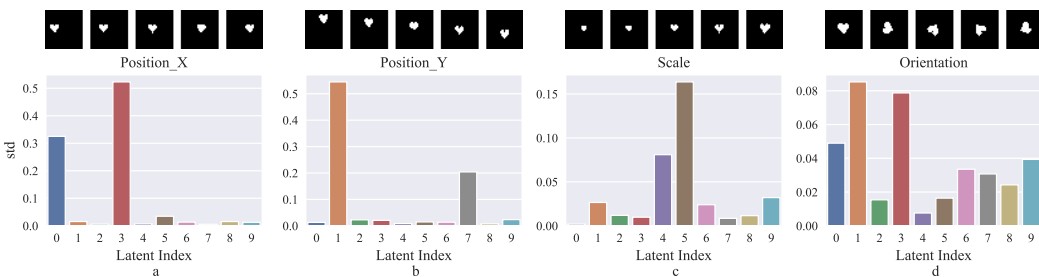

**Figure 3:** Representation variation when manipulating one factor only in the dimension-reduced version. In (a), (b), (c), *position_x*, *position_y* and *scale* are manipulated respectively and only cause one dimension significantly variate. While, in (d), when manipulating the ill-defined factor *orientation*, multiple dimensions variate.

## 4.3 Influence by Manipulating Factors

Another intuitive direction to study the relationship between representations and factors is the influence on representations when manipulating the factors. Given that the original representation vector dimension is much higher than the number of factors, we first make the representation more compact to have a more concise illustration. Here, we reduce the representation dimension by the selection process of top-k MED described in Section 3.3.

Figure 3 shows the result of representation vector variation when changing only one factor at once. We take BYOL on dSprites as a representative. Here we set $k = 2$ for top-k MED, i.e., we pick the 2 most disentangled dimensions for each factor and derive a 10-dim representation. We sample a set of images regarding a specific data factor. These images have all possible values of this data factor while having the same value for all other data factors. We get 10-d representations from these generated images by a shared model. Then, we compute the variance of each of the 10 dimensions across the images, leading to 10 scalars. The larger the variance is, the more that dimension responds to the factor. Figure 3(a), (b) and (c) show how the reduced representation vector changes when manipulating the *position_x*, *position_y*, and *scale* factors respectively. Note that we set $k = 2$, therefore we see good disentanglement, with only exactly two representation dimensions having high variation. However, in Figure 3(d) we show a failure mode of the ill-defined factor *orientation* that the change of factor causes multiple dimensions of reduced representation to have large variations, indicating that this factor is represented in an entangled way. From the results, we observe that manipulating different well-defined independent factors causes evident variance on disjoint sets of dimensions. Further, these results demonstrate the existence of a well-disentangled subset of latent dimensions. We also conduct a similar qualitative study where the representation dimension is reduced by the unsupervised PCA technique and observe a similar pattern. The details are provided in Appendix B.4.

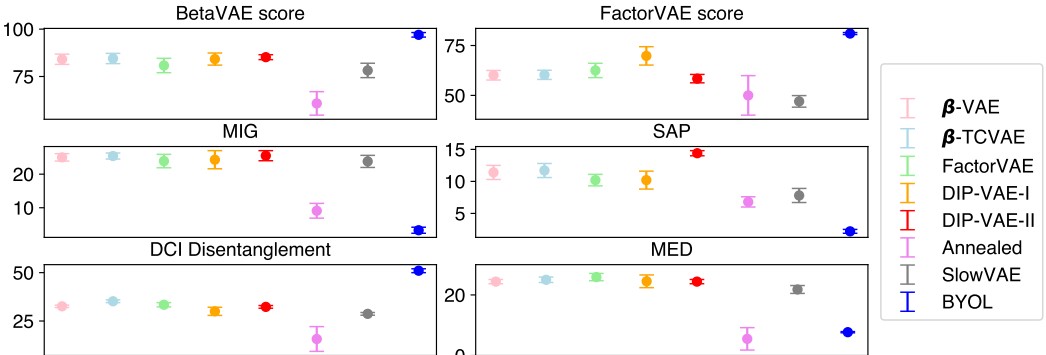

**Figure 4:** Evaluation with multiple metrics on SmallNORB. Y-axis is the corresponding disentanglement scores. The result shows the disagreement among existing metrics in high-dimensional space (BYOL).

# 5 Quantitative Evaluation

In this section, we conduct quantitative studies on the disentanglement property of contrastive learning methods. We first introduce the experiment setup in Section 5.1. Then we show quantitative results under both MED and existing disentanglement metrics in Section 5.2 which shows the disagreement of existing metrics to support the necessity of proposing MED. Finally, we make a full quantitative benchmark of methods of interest with MED in Section 5.3 and an ablation study about the dimension of the representation model in Section 5.5. More quantitative studies are available in Appendix C.

## 5.1 Experiments Setup

The details for reproducibility are introduced in Appendix A. Here we provide a brief description of the setup of the experiments.

**Datasets.** Representation disentanglement is usually evaluated on synthetic datasets, such as dSprites (36), Cars3D (39), Shapes3D (5), and SmallNORB (29). Besides those datasets, we also include a real-world dataset CelebA (32). CelebA contains human face images with 40 binary attributes. The attributes include fine-grained properties of the human face, such as whether wearing glasses or having wavy hair. We include the details of dataset factors in the appendix.

**Evaluation Protocol.** We conduct experiments with both MED and the existing metrics to reveal the disagreement between existing metrics. Then we use MED as the main metric to study the disentanglement of contrastive learning methods. The implementation of evaluation metrics is adapted from the protocol provided by DisLib (34). All results are calculated with three random seeds and we report both the average score and the standard deviation. More details are introduced in Appendix A.2.

**Reference Methods.** We investigate most of the popular disentangled representation learning methods as studied in the standard benchmark of DisLib (34). Besides, we also compare with a recently proposed ICA method called ICE-BeeM (23). Since we do not assume the ground truth factors are known during the training, we use its unconditional version. We term it EBM (energy-based model). For the contrastive learning methods, we evaluate not only negative-free methods such as BYOL (40), Barlow Twins (51) and SimSiam (7), but also those using negative samples such as MoCo and MoCov2 (14).

**Model Implementation.** All methods use a shared architecture of encoder network as explained in the appendix. The latent dimension of contrastive learning methods is set to 1000 since they require a high-dimensional latent space to work. For other methods, the latent dimension is set to be 10 on synthetic datasets as in DisLib and 128 on CelebA dataset when evaluating with MED. For the evaluation with Top-k MED, the dimension of all methods is set to 1000-d for fairness. On dSprites, Cars3D and SmallNORB, we acquire checkpoints from DisLib if they are provided. We train our checkpoints on CelebA and Shapes3D.

## 5.2 Disagreement of Existing Metrics

As there is not a uniform definition of "disentanglement", existing disentanglement metrics are motivated by different desired properties of a disentangled representation. These metrics had previously shown good agreement in the large-scale experiments of DisLib (34). However, when we extend the

disentanglement study beyond low-dimensional scenarios, these metrics disagree significantly. We select representative low-dimensional VAE-based methods and a high-dimensional BYOL model to evaluate on a representative dataset, the SmallNORB dataset. The results are in Figure 4, which show a significant disagreement among metrics on BYOL, while they agree on the low-dimensional VAE methods. Aligned with our analysis in Section 3.2, BetaVAE score, FactorVAE score, and DCI overestimate the disentanglement degree of the high-dimensional model while MIG and SAP underestimate it. Since existing metrics fail to evaluate high-dimensional models, we opt to use MED as the main evaluation metric in the following sections. Please refer to Appendix C and E for more evidence of the disagreement of existing metrics. Moreover, in Appendix F, we provide the analysis by constructing scenarios where MED can still output the result aligned with human intuition while existing metrics fail to conduct a meaningful measurement of disentanglement degree.

**Table 1:** MED and Top-k MED scores on multiple datasets. Methods in gray are contrastive self-supervised learning methods. The latent dimension of VAE methods with * is 1000.

| Metrics | Model | dSprites | Shapes3D | Cars3D | SmallNORB | CelebA |
|---------|-------|----------|----------|--------|-----------|--------|
| **MED** | $\beta$-VAE | 32.6 (10.0) | 52.5 (9.4) | 29.0 (2.2) | 24.4 (0.7) | 3.3 (0.5) |
| | $\beta$-TCVAE | 31.8 (7.4) | 53.2 (4.9) | **33.0 (3.8)** | 25.0 (0.9) | 4.7 (0.1) |
| | FactorVAE | 32.5 (10.1) | 55.9 (8.0) | 29.1 (3.0) | **25.9 (1.2)** | 0.6 (0.6) |
| | DIP-VAE-I | 18.8 (5.6) | 43.5 (3.7) | 19.4 (3.3) | 24.5 (2.1) | 3.7 (0.2) |
| | DIP-VAE-II | 14.7 (5.5) | 52.6 (5.2) | 16.7 (4.1) | 24.4 (0.6) | – |
| | AnnealedVAE | **35.8 (0.8)** | **56.1 (1.5)** | 15.5 (2.5) | 5.5 (3.7) | – |
| | EBM | 6.8 (4.0) | 2.1 (2.6) | – | 2.3 (1.7) | – |
| | MoCo | 4.2 (0.5) | 6.1 (0.1) | 8.6 (0.4) | 4.9 (0.1) | **5.8 (0.1)** |
| | MoCov2 | 3.5 (1.4) | 4.2 (0.4) | 6.5 (0.4) | 3.3 (0.2) | 4.8 (0.2) |
| | BarlowTwins | 6.0 (0.3) | 6.4 (0.3) | 5.6 (1.4) | 6.1 (0.1) | 4.2 (0.2) |
| | SimSiam | 26.5 (0.1) | 12.3 (0.9) | 10.4 (0.1) | 10.7 (1.1) | 5.3 (0.4) |
| | BYOL | 31.3 (0.4) | 6.0 (0.5) | 9.7 (0.5) | 7.7 (0.2) | 4.8 (0.4) |
| **Top-k MED** | $\beta$-VAE* | 16.6 (6.2) | 19.2 (1.4) | 29.2 (2.0) | 15.8 (2.1) | 4.5 (0.3) |
| | $\beta$-TCVAE* | 11.2 (0.4) | 25.0 (0.5) | 20.0 (1.8) | 23.0 (0.6) | 3.6 (0.2) |
| | FactorVAE* | 3.2 (3.8) | 8.2 (4.0) | 7.8 (1.6) | 4.8 (1.0) | 5.0 (0.3) |
| | DIP-VAE-I* | 7.0 (1.3) | 16.2 (0.9) | 24.6 (2.2) | 20.9 (2.7) | 2.5 (0.9) |
| | MoCo | 16.1 (2.0) | 18.1 (0.6) | 26.6 (1.6) | 17.9 (0.8) | **7.9 (0.1)** |
| | MoCov2 | 14.7 (1.0) | 13.6 (1.7) | 24.5 (2.1) | 15.1 (0.9) | 6.6 (0.7) |
| | BarlowTwins | 21.7 (1.3) | 20.0 (0.3) | 23.8 (2.5) | 24.5 (1.5) | 5.7 (0.2) |
| | SimSiam | 39.1 (0.4) | **30.0 (2.0)** | **32.7 (2.3)** | **28.4 (1.9)** | 7.2 (0.6) |
| | BYOL | **53.7 (0.7)** | 19.7 (1.3) | 31.8 (1.3) | 25.7 (0.3) | 6.8 (0.7) |

### 5.3 Disentanglement Benchmark with Contrastive Learning Methods

For the benchmarking of disentanglement, we use both MED and the partial version of MED, i.e. Top-k MED. For Top-k MED, we set the MED partial evaluation hyperparameter $k = 2$ for dSprites, Shape3D, and SmallNORB, and $k = 3$ for Cars3D and CelebA. The values of $k$ are chosen such that the selected dimensions are roughly close to the latent space dimension of the low-dimensional reference methods. And we extend the latent dimension of all methods to 1000 when evaluating Top-k MED for fairness. We note that despite our hyperparameter search, we were unable to train good EBM weights on Cars3D and CelebA, so we keep that section empty. The results are in Table 1. We also encourage readers to read the results of existing metrics in Table 4 in Appendix C.

In the upper part of Table 1, we show the MED score for previous disentangled methods as well as contrastive methods. We find that contrastive methods achieve significantly lower disentanglement scores on 3 of the 5 datasets (Shapes3D, Cars3D, SmallNORB). Contrastive methods achieve slightly higher disentanglement scores on CelebA. On dSprites, some negative-free contrastive methods (SimSiam, BYOL) achieve scores close to SOTA, but the other contrastive methods' score is much lower. In summary, in most cases, contrastive methods have inferior disentanglement properties compared to the best methods; only in a few settings do contrastive methods achieve scores comparable to SOTA scores.

These results are disappointing but not surprising since contrastive methods are not explicitly designed to maximize the feature disentanglement. Further, since the underlying number of factors is usually quite small, on the order of 10, the 1000-d feature space will likely have dimensions that are either not related to the ground truth factors or capture a combination of the ground truth factors. This result

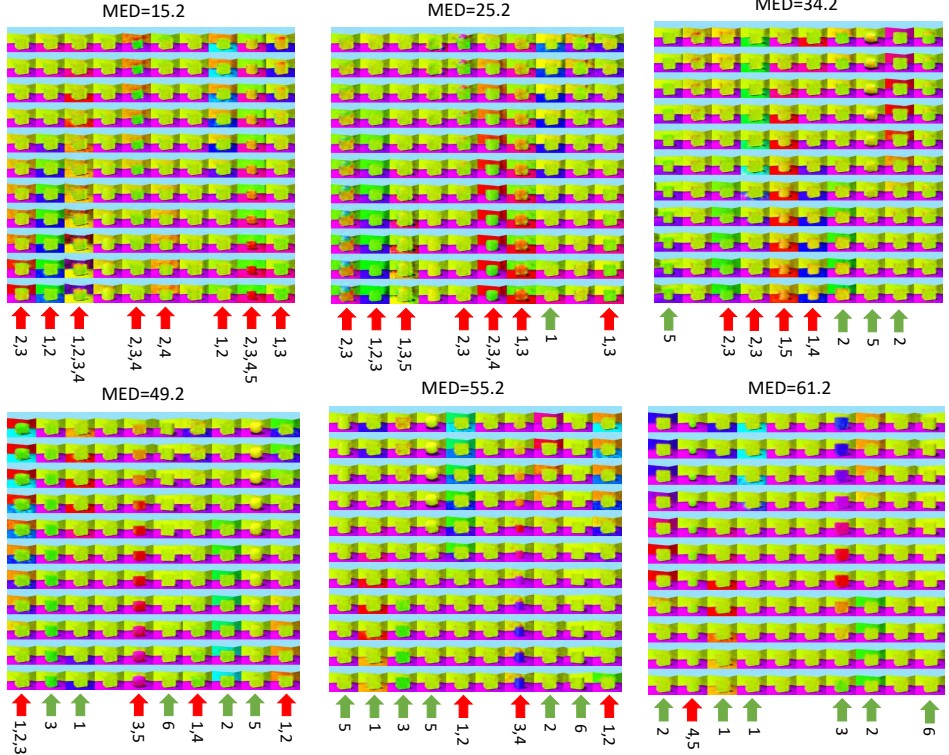

**Figure 5:** The visualization of latent traversing by VAE models from Van Steenkiste et al.. Each subfigure is the visualization for one model, annotated with the MED score of that model. The involved data factors include (1) Floor color; (2) Wall color; (3) Object color; (4) Object scale; (5) Object shape and (6) Orientation. For each column in the subfigure, the value of only one latent dimension changes, i.e. latent traversal for that dimension. Below each column, we also annotate the actual changing factors in this column, obtained with human manual inspection. If a model is fully disentangled, we expect each column to correspond to only one ground truth factor. The visualization shows that if a model achieves a higher MED score, it has more dimensions disentangled and responsive to only one factor while fewer dimensions entangled to multiple factors.

does not contradict with Zimmermann et al. (53) since (1) they find that the learned feature is a linear transformation of the ground truth factors, which doesn't necessarily disentangle, and (2) they use augmentations on factors that cannot be done in practice.

The lower part of Table 1 shows top-k MED measurements on various methods. We find that contrastive learning methods (especially the negative-free ones) in general show a better disentanglement in a selected subspace and the disentanglement is stronger than the reference methods. This shows that there exists a subspace in the learned representation that is well disentangled. Moreover, when we compare the subspace in contrastive methods (gray part in the lower section of Table 5.3) to the traditional approach that directly trains a low-dimensional latent space (non-gray part in the upper section of Table 5.3), we find that the disentanglement of the former is usually better than or on par with that of the latter. This means that we probably should not constrain the dimension of the latent space and require it to be fully disentangled, but rather should use high dimensional latent spaces and only require it to have a subset with good disentanglement properties.

To conclude, we find that the high-dimensional contrastive methods, including negative-free ones, do not learn a fully disentangled representation. However, there exists a subspace in the learned representation that is well disentangled. Such a subspace can show much better disentanglement properties than previous SOTA approaches. Despite the fact that these methods require a high dimension to train, such a subspace can serve as a proxy between contrastive learning methods and a more compact and disentangled low-dimensional representation.

### 5.4 Latent Traverse on Shapes3D

Now, we provide visualizations to show that MED can provide evaluation results aligned with human intuition about the disentanglement degree regarding data factors. Because CL methods

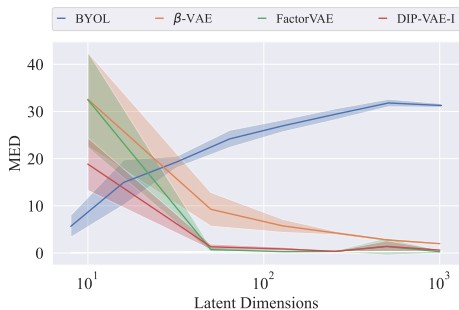

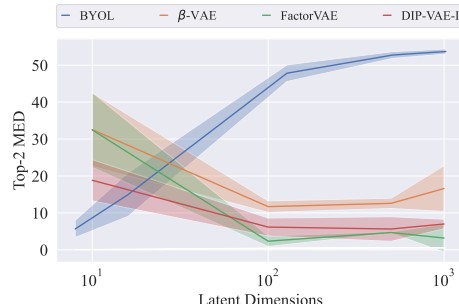

**(a)** MED score w/ dimension change.

**(b)** Top-2 MED score w/ dimension change.

**Figure 6:** The influence of the representation dimension on the (a) MED score and (b) Top-2 MED score on dSprites dataset. The disentanglement property regarding both metrics is enhanced with increasing representation dimension for the BYOL method but decreased for all tested VAE-based methods. The minimum dimension is 10 for VAE-based methods, and 8 for BYOL.

are not generative models and have high dimensions, they are hard to be adopted for visualization with latent traversal. We thus adopt VAE-based generative models here to perform latent traverse. We follow the practice in DisLib (34) to use Shapes3D for the latent traverse. For fairness and reproducibility, we trained VAE models on Shapes3D with the published configurations provided by Van Steenkiste et al. because DisLib (34) does not provide the raw experiment logs on Shapes3D. The results are shown in Figure 5. For each column in each subfigure, only the value of one dimension of the latent code is manipulated. The manipulation is performed the same way as the default setup for traverse visualization in DisLib. Given the six factors on Shapes3D with index, we indicate under each subfigure if (1) red arrow: a dimension is entangled to more than one factor or (2) green arrow: a dimension is disentangled and responsive to only one factor. Through the visualizations and the corresponding MED scores, we can clearly see that MED can well represent the disentanglement degree. We could observe a clear pattern that the model with a higher MED score has more disentangled representations. This demonstrates that the results from MED scores are aligned with the intuition of humans.

### 5.5 Influence of Dimension

As representation dimension is found as a core variable in disentangled representation learning, we evaluate the influence of dimension over disentanglement by both MED and top-k MED on dSprites with BYOL as an example versus VAE-based methods. The results are shown in Figure 6. We find that the BYOL's MED score and top-2 MED score increase along with the latent dimension. The scores plateau at around 512 dimensions. This is consistent with previous literature on the difficulty to train informative contrastive models with low latent dimensions (12). We further show that a lower latent dimension leads to a less disentangled subspace as well. On the contrary, we also note that the VAE methods fail to scale to higher latent dimensions. We find a large gap between higher-dimensional VAEs and their 10-dim versions. This suggests the gap between existing disentangled representation methods and the real-world data complexity, which can not be represented in a limited dimension.

## 6 Conclusion

In this paper, we provide an empirical study of the disentanglement property of contrastive learning without negatives for the first time. In the high-dimensional space, we find the difficulty of adopting the existing disentanglement metrics. Therefore, we propose MED and Top-K MED to evaluate disentanglement based on mutual information. The evaluation shows that even without negative samples, contrastive learning can learn a well-disentangled subset of representation. Recently, the study of contrastive learning, or general self-supervised learning, is still motivated by empirical observations. We hope our work can reveal some clues to motivate future theoretical justifications.

### Acknowledgement

We appreciate the help from Jinhyung Park on paper writing. This work is supported by the Ministry of Science and Technology of the People's Republic of China, the 2030 Innovation Megaprojects "Program on New Generation Artificial Intelligence" (Grant No. 2021AAA0150000). This work is also supported by a grant from the Guoqiang Institute, Tsinghua University.

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
