# OpenReview forum: "An Empirical Study on Disentanglement of Negative-free Contrastive Learning"
_NeurIPS.cc/2022/Conference — NeurIPS 2022 Accept_

### Official Review · Reviewer_s3YA · 2022-06-25

**Rating:** 5
**Confidence:** 4
**Soundness:** 3 good
**Presentation:** 3 good
**Contribution:** 2 fair

**Summary:**

This paper empirically studied the disentanglement property of self-supervised methods, such as MoCo, BarlowTwins, and BYOL. Besides, the authors validated the disagreement of current disentanglement metrics for the models with high-dimensional latent space. The authors proposed a new metric based on mutual information to measure high-dimensional representations. Massive experiments conducted on synthetic datasets and a real-world dataset showed that negative-free contrastive methods can learn a disentangled subset of representation.

Contribution:
1. This work studied the disentanglement properties of negative-free contrastive models for the first time.
2. This work proposed a new metric for high-dimensional models and a selection strategy to pick a disentangled subset of representation.

**Questions:**

The authors claim that disentanglement property remains unexplored in negative-free contrastive learning, but I find some relevant papers, e.g., Contrastive Disentanglement in Generative Adversarial Networks. Could the authors discuss the differences? Why negative-free contrastive learning is important?

How to prove the superiority of your proposed metric? Beta VAE, Factor VAE score, and DCI disentanglement show positive results for BYOL.




**Ethics Review Area:**

["I don’t know"]

**Limitations:**

The authors did not show the superiority of the proposed metric. They need to find some cases where all metrics fail to measure the disentanglement except the proposed metric.



**Strengths And Weaknesses:**

Strength:
1. The experiments were comprehensive and massive to cover popular conventional disentanglement methods and negative-free contrastive models.
2. This paper aims to address the disentanglement measurement of high-dimensional representations and bring negative-free contrastive learning into disentanglement learning.

Weakness:
1. line 34: “latent representation, This” → “latent representation. This”
2. Comparing metrics for a model in a subfigure may better show the disagreement or agreement of metrics in Figure 4.
3. We know Orientation is hard to be disentangled, but the authors still need to show ALL experimental results without selection.
4. What is manipulating a factor in section 4.3? Getting a set of images by traversing one factor? Could you color the latent index with the factor you picked? I cannot see the existence of a well-disentangled subset for orientation.

---

> ### Author Response · Authors · 2022-08-02
> **Response to Reviewer s3YA #1: Paper details**
>
> Dear reviewer,
>
> Thank you so much for your review and suggestions. Below are our responses to your questions.
>
> **Q1: Grammar issues:** We have fixed the issue in the revised version.
>
> **Q2: Suggestions for re-draw Figure 4:** We provide a new version of Figure 4 in the revised draft (**Figure 15**) where the value of multiple metrics is included in a single subfigure. However, we note that including multiple metrics in a single subfigure may cause confusion because the value of different metrics can't be compared directly and they are usually at different scales, so this figure can't well explain the disagreement between these metrics. So instead, we visualize the ranking of the corresponding methods on SmallNORB with different metrics in **Figure 16**. We believe this can clearly show the disagreement of existing methods when the representation dimension varies. Also, we highly encourage to refer to **Table 4**, which shows complete results on different methods and metrics, more convincingly suggesting the disagreement of existing metrics.
>
> **Q3: Include orientation in results on dSprites:** For all the evaluation results, the orientation factor was already included without selection. We excluded it just in the visualization to avoid misleading or confusing. But yes, a full presentation helps deliver a more complete picture. We make a new visualization with orientation factor in **Figure 7(c)** and **Figure 8(d)**.
>
> **Q4: Clarification of factor manipulation.** For questions related here, we answer them by points:
>
> **Q4.1: How the traverse is done?**
>
> **A4.1:** Unlike generative methods, contrastive learning methods do not model the generating process from factors to images. So manipulating factors here means we sample a batch of images from the dataset with only one factor varies. This is possible since we have access to the ground truth factors for synthetic datasets like dSprites. Then we extract representations from the corresponding batch of images for each factor.
>
> **Q4.2. Could you color the latent index with the factor you picked?**
>
> **A4.2:** We have included a new visualization of the top-k process in **Figure 13**,  where the selected dimensions are highlighted in yellow. Please refer to **Appendix F.3**.
>
> **Q4.3: I cannot see the existence of a well-disentangled subset for orientation.**
>
> **Q4.3:** Yes that is true. In **Figure 3**, we show orientation as a failure case. As explained in Section 4.1 and **Appendix B.2** and **B.3**, it is due to the orientation factor being ill-defined.
>
> **Q5: I find some relevant papers, e.g., Contrastive Disentanglement in Generative Adversarial Networks [P1]. Could the authors discuss the differences?**
>
> **A5:** Contrastive learning and method in [P1] are very different from multiple perspectives:
> 1.  As we discussed in the introduction (**L21**), contrastive learning is a class of discriminative methods while previously studied disentangled representation learning methods are generative models. And the method proposed in [P1] is still a generative method (GAN) instead of a contrastive learning method.
> 2.  Second, In [P1], the term "contrastive" is misleading. The “contrastive loss” they use is actually a **"classification"** instead of a **"contrastive"** loss. In the common definition of contrastive learning, "contrast" comes from the feature distance of positive and negative pairs. And the positive pairs come from the same raw data sample. However, in the mentioned paper, the positive/negative pairs are defined as samples from the same/different categories, which is in fact classification instead of feature contrast.
> 3.  Finally, the metrics in [P1] are all for classification/clustering performance (ACC/NMI/NRI) or generation performance (IS/FID). They do not show any evaluation on the feature disentanglement metrics.

---

> ### Author Response · Authors · 2022-08-02
> **Response to Reviewer s3YA #2: Why negative-free CL is important**
>
> **Q6: Why negative-free contrastive learning (CL) is important.**
>
> **A6:** Negative-free CL (also termed as "Non-contrastive Self-supervised Learning") is important for multiple reasons, especially in the comparison with CL with negative samples:
> 1. First, **why self-supervised learning (SSL) is important**: self-supervised learning proposes a path to training large-scale pre-trained models without expensive labeling. The success of a line of this work, such as MoCo, SimCLR, and BYOL, has made the performance on downstream tasks close to or even superior to supervised-pre-trained models.
> 2. Second, **why negative-free contrastive SSL:**
> *  Contrastive SSL often suffers from model collapse that the representation becomes trivial constant vectors. By removing negatives, previous works observed the relief of this problem. This may be the main reason why researchers such as Yann Lecun (https://twitter.com/ylecun/st suffers from model collapse that the representation becomes trivial constant vectors. By removing negatives, previous works observed the relief of this problem. This may be the main reason why researchers such as Yann Lecun (https://twitter.com/ylecun/status/1508253087789637643) puts more attention to negative-free contrastive learning. Many recent works have discussed this, such as
>         1. https://arxiv.org/pdf/2105.00470.pdf (ICCV'2021 oral)
>         2. https://openreview.net/forum?id=r4xe3nMQ3AY  (ICML 2021 Outstanding Paper Honorable Mention)
>         3. https://openreview.net/forum?id=YevsQ05DEN7 (ICLR'2022)
> *   The success of SSL relies on heavy empirical observations and heuristic practice, such as stop-gradient, the set of projector and predictor, moving average update, etc. Thus researchers are always seeking more theoretical understanding of it. Though the original study of SSL uses negative pairs as a necessary component, later research found it is at the core of SSL's good performance. So, by removing negative pairs from SSL, many works make good progress in understanding the internal mechanism of SSL. Related works include:
>         1. https://arxiv.org/pdf/2206.02574.pdf
>         2. https://arxiv.org/pdf/2205.11508.pdf
>         3. https://openreview.net/forum?id=CR1XOQ0UTh- (ICLR'2021)
>         4. https://ieeexplore.ieee.org/document/9706613 (WACV'2022)
> *   Contrastive SSL with negative samples requires heavy computation of data augmentation, and it also introduces additional heuristic choices. This proposes a risk of generalizability on different datasets or tasks. For the details of this aspect, related works include:
>         1. https://arxiv.org/pdf/2002.05709.pdf (ICML'2020)
>         2. https://arxiv.org/abs/2109.05941 (EMNLP'2021)
>         3. https://arxiv.org/pdf/2007.13916.pdf (NeurIPS'2020)
>
> Given all the discussion above, considering (1) the performance of pretraining by negative-free contrastive learning can already be comparable to traditional contrastive learning, (2) helping to understand SSL, (3) avoiding model collapse, or (4) more generalizable pretraining, it is important to study for more understanding of negative-free contrastive learning. And disentanglement property is one of the core characteristics to understand it.

---

> ### Author Response · Authors · 2022-08-02
> **Response to Reviewer s3YA #3: Superiority of MED**
>
> **Q7: The superiority of MED**
>
> **A7 <The response is partially duplicated from that to Q2 of Reviewer beE6>:**  As we discussed in **Section 3.2** and the newly added **Appendix F.2**, the positive results from traditional metrics such as BetaVAE, FactorVAE, and DCI can NOT prove that the BYOL method has overall better disentanglement because those metrics are not applicable to high dimensional cases.
> 1.  The BetaVAE and FactorVAE scores develop classifiers to identify a dimension that correlates most to the intervening factor. Higher dimensional representations with redundant information make it easy to overfit the classifiers, leading to higher final scores. For example, in **L137, "a randomly initialized 1000-d model could reach a FactorVAE score of 61.4 on dSprites, close to many well-trained 10-d VAE-based models’ scores"**.
> 2.   MIG and SAP require an extra property, i.e., a factor should be responded to by only a single dimension, termed **completeness**. Moreover, the significant difference between representation dimensions and the number of factors (1000 v.s. 10) enlarges the disagreement between **disentanglement** and **completeness**.
> 3.   The reason for the traditional DCI metric's failure is that it uses a classifier, such as GBT, that gives an unfair advantage to high-dimensional representations. GBT encourages sparsity (see **Figure 6**), leading to underestimating some responses between factors and latent dimensions. With a dimension correlating with fewer factors, it scores high-dimensional dimensions **falsely high**. Or at least, we can not use the score to compare with other methods of different dimensions.
> 4.   Our metric MED is a fix based on the traditional DCI metric. Instead of GBT, we use MI per dimension, which won't give extra advantages to the high dimensional representation. MED only focuses on disentanglement measured by the entropy of importance matrix rows, decoupled from completeness (which should be the entropy of columns). In addition, our empirical justification of the MED metric show MED is a plausible metric for the high-dimensional case. Please refer to **Appendix F.2** for a detailed discussion.
> 5.  The newly added **Appendix G** also provide a lot of evidence. For example, **Figure 16** shows that the traditional metrics are usually consistent when ranking a low-dimensional model. But for BYOL, MIG and SAP score it as the worst, but betaVAE, factorVAE, and DCI score it as the best. The MED score, on the other hand, is designed to be able to handle the high dimensional latent space.

---

> > ### Comment · Reviewer_s3YA · 2022-08-04
> > **MED probably measures disentanglement, but not convincedly.**
> >
> > I am not doubting the advantages of the proposed metric. Of course, MED measures a property, but there is a gap in proving that such a property is disentanglement. You need a justified third party. For example, you can use latent traversals to visualize latent variables in low-dimension cases. A trustable third party is required to show superiority.

---

> > > ### Author Response · Authors · 2022-08-06
> > > **Visualization of traversing latent as the third party**
> > >
> > > Thanks for the reply and suggestions.
> > >
> > > We update the draft and add a new section **Appendix H**. In this part, we do latent traversing of low-dimensional (10-d) models on Shapes3D for visualization in **Figure 17**. We select 6 model weights whose MED scores are divergent. We make a subfigure for each model. Each subfigure has 10 columns, at each of which we change the value of a single latent dimension. In such a manner, we could observe if a latent dimension is correlated to a factor or not.
> > >
> > > By manual inspection, we observe which factors each latent dimension is correlated to. We use a red arrow to indicate a latent dimension if we find it is entangled to express multiple factors. We use a green arrow to indicate a latent dimension if it is disentangled to correlate to only one factor. And we comment on the indices of factors the latent dimensions are correlated to.
> > >
> > > Such a latent traversing visualization follows the practice in DisLib. Through the visualization, we could find that if a model has a higher MED score, it is obvious that it has more dimensions disentangled to represent only one factor, while fewer dimensions entangled to multiple factors. I think this can serve as a convening third party to show the consistency between MED score and the intuition of dimension-factor disentanglement by human sense.

---

> > ### Comment · Reviewer_s3YA · 2022-08-07
> > **Why contrastive learning works?**
> >
> > I have a hypothesis about the poor performance of CSL in low dimensions, that is, VAEs naturally encourage compactness of latents. In contrast, the learned representations by CSL have redundant information, causing the model entangled. Maybe each factor need more dimensions to be represented. So, a problem emerges if that is true. If we need 512-d for 5 factors, do we need 5120-d for 50 factors?

---

> > > ### Author Response · Authors · 2022-08-08
> > > **Bottleneck of CL's disentanglement on complicated datasets is not dimension**
> > >
> > > Thanks for the question!
> > >
> > > **According to our experimental results, the answer is: No, we don't need that many dimensions for 50 factors, where the bottleneck is not the dimension**.
> > >
> > > We study the relationship between dimension and disentanglement of BYOL in **Figure 5** on dSprites with has only 5 factors. BYOL's performance is better as we increase the latent dimension, until 512-d. Its performance plateaus near 512 dim (see **Section 5.4**). Beyond that, there is no further benefit by increasing the dimensions. On the CelebA dataset with nearly 10 times more factors (40 factors), this observation keeps almost the same. The following table shows MED and top-3 MED scores of BYOL with variant dimensions on CelebA. We can observe that BYOL's disentanglement performance plateaus at 1024-d. Further increasing dimensions does not obtain advantages. Our experimental results on CelebA throughout the paper are all obtained with 1000-d latent space, and it achieves SoTA disentanglement performance. CelebA is such a challenging real-world dataset that the disentanglement performance on it is overall very low. A higher latent dimension will make no difference.
> > >
> > > |           | 128-d | 256-d | 512-d | 1024-d  | 2048-d  | 4096-d|
> > > |-----------|-------|-------|-------|---------|---------|-------|
> > > | MED       |  3.5 (2.1) | 4.7 (0.2) | 4.6 (0.2) | 4.8 (0.4) | 4.5 (0.2) | 4.4 (0.3) |
> > > | Top-3 MED |  3.9 (2.3) | 5.3 (0.1) | 5.6 (0.1) | 6.8 (0.7) | 6.1 (0.5) | 6.5 (0.5) |
> > >
> > > **The high dimension required by CL is more likely because of its intrinsic design and architecture, such as the redundancy required by the InfoMax principle[1]**. When dimension is lower than a threshold (e.g., around 256-d for BYOL), CL is difficult to have smooth training loss descent and suffers from bad downstream performance. Such an observation happens on datasets of different complexities.  As long as the latent dimension is higher than the factor number and enough for the intrinsic requirement of CL, higher dimensions show no difference. We didn't observe the sensitivity of the dimension with respect to the factor number of datasets.
> > >
> > > References:
> > >
> > > [1] "On Mutual Information Maximization for Representation", ICLR'2020

---

> > ### Author Response · Authors · 2022-08-09
> > **More experimental and theoretical evidence to show MED's superiority**
> >
> > Dear Reviewer s3YA,
> >
> > As stated in the discussion with Reviewer beE6, regarding your concern that we  did not show the superiority of MED, we have added more evidence to demonstrate its superiority. In the revised draft, we added **Appendix I**, which contains both experimental and theoretical analysis to show that in some cases, all other metrics fail to give meaningful and fair results but MED is robust to all situations.
> >
> > You could refer to the discussion with Reviewer beE6 and **Appendix I** in the revised draft for details. Here, we provide a brief description to send the major messages.
> >
> > **1. Experimental Observations:**
> >
> > |                                   | BetaVAE score | FactorVAE score | DCI  | MED  |
> > |-----------------------------------|---------------|-----------------|------|------|
> > | 10-d trained model (DIP-VAE-II)   | 81.5          | 58.6            | 12.3 | **14.7** |
> > | 1000-d randomly initialized model | **82.7**          | **61.4**            | **19.8** | 3.8  |
> >
> > |                                 | MIG | SAP | MED |
> > |---------------------------------|-----|-----|-----|
> > | 10-d randomly initialized model | **6.7** | **3.0** | 2.9 |
> > | 1000-d trained model (BYOL)     | 5.2 | 2.8 | **6.0** |
> >
> >
> > For the disentangled representation learning, if a metric is robust and valid, it is expected not to score high to randomly initialized encoder weights. However, as shown in the two Tables above, because of the bias of existing metrics towards models of different dimensions, **a randomly initialized 1000-d model can achieve higher score than a well-trained 10-d disentangled representation learning method in terms of BetaVAE score / FactorVAE score and DCI. On the other hand, a randomly initialized 10-d model can get higher score than a 1000-d well-trained model in terms of SAP and MIG metrics.** Such an experimental observation obviously counters our expectation and high-level intuition about a robust disentanglement metric. On the contrary, it both cases, MED maintains reasonable scoring to the models.
> >
> > **2. Theoretical Analysis:**
> >
> > In **Appendix I**, we prove case by case to show that existing metrics can fail in many obvious situations but MED is robust in all cases to output meaningful and consistent disentanglement evaluation.
> >
> > 1. **MIG and SAP** biasedly prefer models of low dimensions: MIG and SAP desire not just **disentanglement** but also **completeness** that one factor should relate to only one dimension. With such an intrinsic bias, MIG/SAP prefers models whose dimension is close to the number of factor. We show an example in **Appendix I** where a model's every dimension is perfectly disentangled to a factor but the model dimension is higher than the number of factor. We will get **MIG=SAP=0**, indicating "fully entangled". But in fact, every dimension of the model is perfectly responsive and disentangled to only one factor.
> >
> > 2. **DCI** suffers from the curse of dimensionality: by increasing the dimension of a model, we can encourage DCI to falsely go higher. For example, given a model of $D$ dimensions, of which the first two dimensions are disentangled and all following dimensions are fully entangled to factors, DCI score goes higher with increasing $D$ and finally approaches 100, indicating "perfectly disentangled". This is obviously false and against our expectation.
> >
> > 3. **BetaVAE and FactorVAE score** also prefer models of high dimensions. We provide two situations as example in Appendix I.
> >
> > - In the first situation, all dimensions of the model are entangled to more than one factors. However, we show that, once there is a dimension that has lower response to one dimension than all other dimensions, BetaVAE and FactorVAE score can go as high as 100, indicating "perfectly disentangled". This is obviously a failure of the metrics as in fact every single dimension is correlated to all factors. Biased to such a "lucky" dimension, the two metrics tend to score high to high-dimensional models.
> >
> > - In the second situation (same as the example for DCI), only the first two dimensions are disentangled to the factors while all other dimensions are fully entangled to all factors. BetaVAE and FactorVAE scores would still output 100 to suggest the model is "perfectly disentangled". This counters our expectation to a reasonable disentanglement metric again.
> >
> > Given all the situations above where existing metrics fail to give meaningful and fair scoring, MED is robust to make consistent disentanglement evaluation. The score by MED is well aligned with the human sense and high-level intuition about representation disentanglement.
> >
> > Please refer to the details in **Appendix I** for more complete experiment analysis and mathematical demosntrations.

---

> ### Author Response · Authors · 2022-08-09
> **Do our responses answer your concerns and questions?**
>
> Dear Reviewer s3YA,
>
> As it is close to the end of the author-reviewer discussion stage, we would like to ask if our responses and the follow-up discussions have addressed your concerns and questions adequately. We would be more than happy to discuss further if you still have concerns to any part.
>
> Regards,
>
> The authors.

---

### Official Review · Reviewer_beE6 · 2022-07-08

**Rating:** 5
**Confidence:** 3
**Soundness:** 3 good
**Presentation:** 3 good
**Contribution:** 2 fair

**Summary:**

The paper tries to empirically study the disentanglement of negative sample free contrastive learning methods, e.g. BYOL, SimSiam. The authors find that the existing disentanglement metric does not fit with the disentanglement of high-dimension feature representation space. Thus, the authors propose a new running time-efficient metric named “Mutual information based Entroy Disentanglement” and MED for short. The authors evaluate the new metric on some popular synthetic datasets and a real-world dataset CelebA and argue that negative sample free contrastive learning methods can learn a well-disentangled subset of representation.

**Questions:**

1. The paper only reports performance on CelebA. Why does not evaluate MED on ImageNet?
2. In line 155, it is easy to get ground truth factors for synthetic datasets. How to get ground truth factors for real datasets?
3. In Equation (1), how to calculate Mutual Infomation for two scalars, or do they represent a distribution over the whole dataset?
4. For Partial Disentanglement Evaluation Metric, how to know which k should be used? Binary search?
5. In line 324, how to get k=3 and how to know the latent space dimension of a real dataset?


**Limitations:**

Limitations are the theoretical analysis mentioned above.

**Strengths And Weaknesses:**

Strength:
1. The paper points out previous methods’ drawback and propose a time-efficient metric that is designed for high-dimension space.
2. The author report various ablation studies to show the properties and effectiveness of the new metric, i.e. uniqueness of factor-representation correspondence, and influence from manipulating factors.

Weakness:
1. The main weakness is that the authors do not theoretically prove the new metric is sound. The major message of this paper is that the previous disentanglement metrics are not good enough and the proposed new metric can fix these problems. However, the paper only showed the drawback of previous metrics and for the new metric, the authors only empirically show it may work e.g. Figure 4. Why should we believe the new metric is better than other methods? Why the value of MED can be regarded as a measure of disentanglement property? From my perspective, the best way to prove a proposed metric work is theoretical analysis, i.e. [1], and the paper misses that part. I recommend the authors consider a high-dimension linear case and prove that MED can beat some other metrics or prove some property of MED.
2. Also, in Section 3.2, the high-level intuition of MDE is not clear. The metric has three parts, Equation (1) (2) (3) and it is quite complicated. I did not get a clear insight, i.e., $R_{ij}$ used in $\rho_i$ and $S_i$. Why should we combine them in the way in Equation (3)? The last paragraph of Section 3.2 should be longer and provide more explanation.
3. The authors argue that contrastive learning can learn a well-disentangled subset of representation without negative samples. The conclusion here is weak. There are many further important questions that the paper does not give answers to. Why the model can learn a well-disentangled subset of representation without negative samples? What is the disentanglement difference between methods with and without negative samples? Is there any high-level intuition or further suggestion about this conclusion? More discussion is needed here.

[1] Kornblith, Simon, et al. "Similarity of neural network representations revisited." International Conference on Machine Learning. PMLR, 2019.

---

> ### Author Response · Authors · 2022-08-02
> **Response to Reviewer beE6 #1: The intuition and process details of MED**
>
> Dear reviewer,
>
> Thank you so much for your review and suggestions. Below are our responses to your questions.
>
> **Q1: The intuition and process details of MED.**
>
> **A1:** Thank you for pointing out the concerns about the intuition and soundness of MED.  Besides the discussion in **Section 3.2** about why we need MED and how it makes evaluation, we also provide a more detailed discussion of MED in **Appendix F**. Here we will summarize our response by points.
> 1. **High-level intuition of MED:** We note there is no uniform definition of disentanglement in this community yet which is the reason why many different metrics are designed. But they follow a shared basic understanding of disentanglement: **the degree to which a representation dimension only responds to a single factor**. However, how to measure the "response" or "importance" remains unclear. Some metrics (MIG) use statistical measurement such as mutual information while some others (DCI, BetaVAE, FactorVAE, SAP) use estimates from learnable modules. The foundation that researchers can use them at the same time is DisLib (L1) proves their agreement with each other with large-scale empirical studies. However, per the discussion in **Section 3.2** and the experiment results shown in **Section 5.2** and **Appendix C, G**, they can not agree with each other anymore when the dimension of models varies. More importantly, the scores from those metrics can not be thought fair and meaningful anymore. So, we design MED to avoid their internal bias triggered by the high dimension. Based on the fact that mutual information is a widely recognized tool to estimate factor correlation, we design MED to gain fair and meaningful disentanglement even with varying representation dimensions as explained in **Section 3.2**. And different from MIG, MED evaluates in a global view of all dimensions. This makes MED more meaningful given different model dimensions.
> 2. **Details of MED:** The process of MED is a variant of DCI Disentanglement score in [E1]. It consists of 3 steps:
>    * The first step is to calculate the importance matrix $R$. Its element $R_{ij}$ describes the amount of information of factor $v_j
> $ captured by the $i$-th dimension of learned representation $c_i$, which is defined by their mutual information normalized by the column $R_j$ (see Eq 1).  The normalization ensures the summation of $R_{\cdot j}$ equals 1 to eliminate the effects of difference in the entropy of factors. **Figure 1** is the visualization of the transpose of the importance matrix.
>     * Then we assign a score $S_i$ for each latent dimension to indicate their disentanglement property. Intuitively, if $c_i$ is disentangled, the row $R_i$ will have one element much larger than the others, corresponding to a column with a single bright entry in **Figure 1**. Here we treat each row $R_{i\cdot}$ as a distribution $P_{i\cdot}$ and employ entropy to quantify the intuition above (see Eq 2). To verify the equation, for example, a perfectly disentangled latent dimension, $c_i$,  corresponds to a distribution that $P_{ik}=1, P_{i,j\neq k}=0$, yielding that $S_i = 1$. On the contrary, a totally entangled latent dimension, $c_{i'}$, corresponds to a uniform distribution $P_{i' \cdot}$, yielding that $S_{i'}=1-\log K$, where K is the number of factors.
>      * Finally, Eq 3 is a weighted sum of $S_i$s summarizing the overall disentangle property. The weight $\rho_i$ indicates the relative importance of each latent dimension. Note that in **Figure 1** there are some columns with no bright entries and they will be down-weighted by $\rho_i$. **Combining $\rho_i$ and $R_{ij}$ in Eq 3 is to account for those dead or irrelevant dimensions that encode no or very little information**.

---

> ### Author Response · Authors · 2022-08-02
> **Response to Reviewer beE6 #2: Theoretical analysis and soundness of MED**
>
> **Q2: Theoretical analysis and soundness of MED.**
>
> **A2:**
> * It is tempting to have theoretical guarantees to show some metrics are better than the others. However, while it is possible to provide a theory for some method under a specific metric, it is usually very hard to have a theory for comparing two metrics. This is because we are designing the metrics itself, thus we don't have anything quantitative to measure the metric. Take metrics in classification problems as an example. We can define accuracy as metric A, and F1-score as metric B. It is hard to theoretically prove that metric B is better than metric A. What people do is to show some cases where accuracy fails to capture the quality of a classifier in an imbalanced classification problem. In the meantime, people also show that F1-score can distinguish the quality of two classifiers.
> * MED is meaningful because MI and Entropy are widely recognized tools to measure variable correlations. What we did for the disentanglement metric is similar to the classification metric above: we show that in the cases we are interested in, the traditional metrics fail (**Section 3.2, Appendix F.2, and Appendix G**). We analyzed the reasons for the failure.
>     *  The BetaVAE and FactorVAE scores develop classifiers to identify a dimension that correlates most to the intervening factor. Higher dimensional representations with redundant information make it easy to overfit the classifiers, leading to higher final scores. For example, in **L137, "a randomly initialized 1000-d model could reach a FactorVAE score of 61.4 on dSprites, close to many well-trained 10-d VAE-based models’ scores"**.
>     *  MIG and SAP require an extra property, i.e., a factor should be responded to by only a single dimension, termed as **completeness**. Moreover, the significant difference between representation dimensions and the number of factors (1000 v.s. 10) enlarges the disagreement between **disentanglement** and **completeness**.
>     *  The reason for the traditional DCI metric's failure is that it uses a GBT classifier and that gives an unfair advantage to high-dimensional representations. GBT encourages sparsity (see **Figure 6**), leading to underestimating some responses between factors and latent dimensions. With a dimension correlates with fewer factors, it scores high-dimensional dimensions falsely high.
>     *  Our metric MED is a fix based on the traditional DCI metric. Instead of GBT, we use MI per dimension, which won't give extra advantages to the high dimensional representation. MED only focuses on disentanglement measured by the entropy of importance matrix rows, decoupled from completeness (which should be the entropy of columns). In addition, our empirical justification of the MED metric show MED is a plausible metric for the high-dimensional case. Please refer to **Appendix F.2** for a detailed discussion.

---

> > ### Comment · Reviewer_beE6 · 2022-08-05
> > **Response to Authors**
> >
> > I appreciate the authors' thorough revision and detailed response. I agree that the previous metric has some problems. However, I cannot draw the conclusion that MED is a good metric for disentanglement. For example, from Table 1 and Table 4, how can we say MED is a good metric? I cannot see a clear relationship here. I agree with Reviewer s3YA. We need a third party to verify the metric is aligned with disentanglement.
> >
> > For the F1-score and accuracy case, we can build different synthetic scenarios and theoretically analyze them. Then we can draw a conclusion that "accuracy fails to capture the quality of a classifier in an imbalanced classification problem but F1 score can capture it". I would like to see a similar conclusion for MED, e.g., the MED is better than other metrics in some properties under some conditions or scenarios.

---

> > > ### Author Response · Authors · 2022-08-06
> > > **2. More evidence of MED's superiority [Part II: Theoretical Analysis]**
> > >
> > > 2.2 **Theoretical Analysis**: Following your suggestions, we add additional theoretical analysis to compare MED and existing metrics in our constructed synthetic linear cases. For the discussions, we assume there are two factors $v_0, v_1 \in {0,1}$. And their value is determined by uniform chances, i.e. $p(v_j=1) = p(v_j=1)=0.5, j=0,1$. The latent representation $c$ is of $D$ dimensions. The details are in **Appendix I**.
> > >
> > > *Note:* By the definition, MED is a percentage number (its maximum value is 1 = 100%), which is the same as other metrics. For the analysis below, we don't write % to simplify the expression and align that with the convention in paper.
> > >
> > > (1) **MED vs MIG/SAP**: In the linear case we construct where each dimension of the model is perfectly disentangled to a factor: $c_i = v_{i mode 2}$. Through the human sense of disentanglement, we could expect a reasonable metric to output a high score in this situation. But in fact, we will get MIG(c) = SAP(c) =0, indicating "totally entangled"! On the other hand, MED(c)=1, indicating a good disentanglement as we expect and derive from intuition.
> > >
> > > (2) **MED vs DCI**: In the linear case we construct, we see that DCI: We construct a linear situation as
> > > \begin{equation}
> > > c_i = \\begin{cases}v_0 &,i=0\\\\
> > > v_1 &,i=1\\\\
> > > 0.5(v_0+v_1) &,i>1\\\\
> > > \end{cases}
> > > \end{equation}
> > >
> > > And we simplify the case to assume that the regressor, such as GBT, in DCI uses the absolute of first-derivative as the importance of a dimension with respect to a factor. Under the sparsity limit of GBT, we could see that the expectation of DCI score is
> > > $$1-\frac{log2}{D-2},$$
> > >  and MED score is
> > > $$1-\frac{D-2}{D}log2.$$
> > > We plot their value change with the variance of the dimension $D$ in **Figure 18**. We see that even if only the first two dimensions are disentangled and all following dimensions are fully entangled, DCI score increases along with increasing $D$ by adding more entangled dimensions. We have $DCI(c)=69.9$ for $D=3$ and $DCI(c)=99.9$ for $D=1000$. This is significantly not aligned with our high-level expectation to a good disentanglement metric. On the other hand, we have $MED(c)=76.9$ for $D=3$ and $MED(c)=30.8$ for $D=1000$, which is more reasonable: with the increase of total dimension, the overall disentanglement degree decreases.
> > >
> > > (3) **MED vs BetaVAE/FactorVAE**: We construct two situations to show how MED is better than BetaVAE/FactorVAE scores.
> > >
> > > **Situation 1:**   We construct a situation as
> > >
> > > \begin{equation}
> > > c_i = \begin{cases} \frac{1}{3}v_0 + \frac{2}{3}v_1 &, i=0\\\\
> > > \frac{1}{3}v_1 + \frac{2}{3}v_0 &, i=1\\\\
> > > \frac{1}{2}(v_0+v_1) &,i>1\end{cases}
> > > \end{equation}
> > >
> > > This is a very entangled situation in which every dimension is correlated to all factors. But by the calculation of BetaVAE and FactorVAE scores, they can achieve 100 for this case, falsely indicating "perfectly disentangled". This is obviously not aligned with the human sense and intuition. On the other hand, we have $MED(c)=1 - log2 = 30.7$, indicating a relatively low degree of disentanglement. We would regard the score from MEDa as still valid while BetaVAE/FactorVAE scores fail significantly.
> > >
> > > **Situation 2:** For the situation same as we construct in the comparison with DCI where the first two dimensions are perfectly disentangled to the two factors respectively while all following dimensions are full entangled, BetaVAE and FactorVAE scores would still give 100 scores, indicating "perfectly disentangled". While for the case of MED, we have $MED(c)=30.8$, which can better reflect the high-level justification that only limited dimensions are disentangled and most dimensions are highly entangled. Moreover, in this situation, by selecting the suitable subset, we can achieve a high Top-k MED score. This shows the meaning of the Top-k MED score especially when the model dimension is higher than the number of underlying factors.
> > >
> > > With all the situations constructed and analyzed above, we could notice the failure of existing metrics to achieve evaluation results (1) meaningful, (2) fair, and (3) aligned with the human sense to disentanglement in some scenarios. On the other hand, our proposed MED always keeps the evaluation and comparison reasonable and aligned with the expectation from the human high-level institution.

---

> > > ### Author Response · Authors · 2022-08-06
> > > **2. More evidence of MED's superiority [Part I: More Experimental Observations]**
> > >
> > > |                                   | BetaVAE score | FactorVAE score | DCI  | MED  |
> > > |-----------------------------------|---------------|-----------------|------|------|
> > > | 10-d trained model (DIP-VAE-II)   | 81.5          | 58.6            | 12.3 | **14.7** |
> > > | 1000-d randomly initialized model | **82.7**          | **61.4**            | **19.8** | 3.8  |
> > >
> > > |                                 | MIG | SAP | MED |
> > > |---------------------------------|-----|-----|-----|
> > > | 10-d randomly initialized model | **6.7** | **3.0** | 2.9 |
> > > | 1000-d trained model (BYOL)     | 5.2 | 2.8 | **6.0** |
> > >
> > > Your suggestions are great. We add more evidence to show that generally or in some cases, MED is better than other metrics by keeping a consistent evaluation of method disentanglement. We add **Appendix I** for this part from both an experimental perspective and a theoretical perspective.
> > >
> > > 2.1 **Experimental Evidence**: By intuition and qualitative understanding of representation disentanglement, randomly initialized weights should not show high disentanglement score with a meaningful metric. But we extend the observation in L136 (**"a randomly initialized 1000-d model could reach a FactorVAE score of 61.4 on dSprites, close to many well-trained 10-d VAE-based models’ scores"**.) in Appendix I.1 with more detailed and comprehensive experimental observations. Results are shown in the two Tables above.
> > >
> > > The results that, on dSprites, the BetaVAE score/FactorVAE score/DCI tend to overestimate the disentanglement property of the model of higher dimensions as we analyzed from their definition in Section 3.2 and Appendix A. To be precise, a randomly initialized 1000-d model can achieve a higher score by these metrics than a well-trained 10-d model. This is obviously not aligned with our expectation to a metric robust to the dimension variance in disentanglement measurement.
> > >
> > > On the other hand, in the second Table on Shapes3D, we see that MIG/SAP tend to underestimate the disentanglement of the model of higher dimensions. Even a randomly initialized 10-d model can get a higher disentanglement score by MIG and SAP than a trained 1000-d trained model. The reason is that SAP and MIG use only two dimensions to calculate the score and they desire additional completeness other than disentanglement. So they tend to overestimate methods' disentanglement if their dimension number is close to the factor number.

---

> > > ### Author Response · Authors · 2022-08-06
> > > **1. Add third party show that MED is aligned with human understanding of disentanglement**
> > >
> > > Thank you for the follow-up discussions.  For your questions:
> > >
> > > 1.**Additional third party to verify MED is aligned with disentanglement**: In Table 1, we provide the comparison between CL methods and existing methods by the proposed new metrics. In Table 4, we want to show that, with the participation of high-dimensional methods, existing metrics can not agree with each other anymore. And your suggestion is good, so we add additional visualization as third party to verify MED is aligned with the human sense of disentanglement and the shared definition: to which degree a dimension is related to only one factor.
> > >
> > > *<The following content is duplicated with that to Reviewer s3YA>*
> > >
> > > We update the draft and add a new section **Appendix H**. In this part, we do latent traversing of low-dimensional (10-d) models on Shapes3D for visualization (**Figure 17**). We select 6 model weights whose MED scores are divergent. We make a subfigure for each model. Each subfigure has 10 columns, at each of which we change the value of a single latent dimension. In such a manner, we could observe if a latent dimension is correlated to a factor or not.
> > >
> > > By manual inspection, we observe which factors each latent dimension is correlated to. We use a red arrow to indicate a latent dimension if we find it is entangled to express multiple factors. We use a green arrow to indicate a latent dimension if it is disentangled to correlate to only one factor. And we comment on the indices of factors the latent dimensions are correlated to.
> > >
> > > Such a latent traversing visualization follows the practice in DisLib. Through the visualization, we could find that if a model has a higher MED score, it is obvious that it has more dimensions disentangled to represent only one factor, while fewer dimensions entangled to multiple factors. I think this can serve as a convening third party to show the consistency between MED score and the intuition of dimension-factor disentanglement by human sense.

---

> ### Author Response · Authors · 2022-08-02
> **Response to Reviewer beE6 #3: (1) Further discussion of paper conclusion and (2) result on Imagenet?**
>
> **Q3. Further discussion:** in the area of self-supervised learning, most advances are made by empirical observations, such as the setup of moving average update[H1], projector[C1] and predictor[G1], stop-gradient[C2], etc. People have been studying the mechanism of SSL without negative samples, but the progress on this topic is still at an early stage. For example, until recently, the work [T1] analyzes why SSL without negative on linear networks won't collapse to the identity function. We emphasize that our contribution is mainly finding this new phenomenon empirically. In the following, we give some initial hypotheses to why this happens.
>
> **Q3.1: Why the model can learn a well-disentangled subset of representation without negative samples?**
>
> **A3.1:**   Here is our hypothesis. In contrastive learning, the training algorithm pulls the embedding of two augmentations of the single image close to each other and pushes embeddings of different images further apart. Since the embedding space we use in contrastive learning is usually very high dimensional (typically at least 1000-dim), and we need to pull close the two augmentations of the same image, it is intuitively better for the augmentation to only affect a few dimensions of the embedding, instead of all dimensions of the embedding. Because the former would lead to a smaller distance between the two augmented embedding vectors. Thus, this is the intuition why it would encourage disentanglement.
>
> **Q3.2: Disentanglement difference from CL with and without negative samples**
>
> **A3.2:** First of all, the disentanglement property of CL with negatives is also barely explored. In [Z1], researchers find negative samples can encourage identifiability of representation from CL, which is related to disentanglement but not the same as discussed in related works. The finding is based on the assumption of [W1] where (approximately) negative samples should be available in CL. However, negative-free contrastive learning does not satisfy the assumption. Moreover, to the best of our knowledge, we are also the first to benchmark CL methods (not just negative-free ones) on standard datasets (**Table 1** and **Table 4**) for which the proposed MED metric is the necessary tool. So to conclude, before this work, people[W1, Z1] found that negative samples can encourage CL to learn representation with some extent of disentanglement, but no standard evaluation was provided. In this work, we find that CL can learn disentangled representation even without any negative samples and in a novel pattern. We also benchmark them with the other methods on standard datasets.
>
> **Q4: why not provide results on ImageNet?**
>
> **A4:** To evaluate disentanglement on a dataset, we need to know the multiple attributes of an image. For example, in CelebA, we know the person's eye color, nose size, facial color etc. It is not suffice to only know an object label for each image as in ImageNet. Since there is no way to tell whether a representation dimension is entangled on unlabeled factors such as background color, etc, which will serve as noise when studying disentanglement. With the too high-level and limited labels on ImageNet, we can't do factor-controlled evaluation on as its label (category) can not represent all the variance between samples. For example, two images of the same label, e.g., dog, can have totally different background, surroundings etc. CelebA is the only real-world dataset we find that has well-defined and complete factor labels.

---

> ### Author Response · Authors · 2022-08-02
> **Response to Reviewer beE6 #4: Other questions and references**
>
> **Q5. how to get ground truth factor on a real-world dataset.**
>
> **A5:** To have confident analysis of disentanglement evaluation, the factors definition should be conclusive that images with the same value of all factors should be exactly the same. Or deep models can capture other undefined signals to learn discriminative features. Hence, it is extremely difficult to define ground truth factors on real-world datasets. This is a main obstacle preventing disentangled representation learning from extending to real-world datasets. And CelebA is the only real-world dataset we find that has well-controlled factor definitions. Other datasets, such as COCO or Imagenet, only have limited high-level attribution labels, such as object categories, which is far away from the requirement of a factor-controlled disentanglement study.
>
> **Q6. How to calculate Mutual Infomation for two scalars.**
>
> **A6:** As explained at the end of **Appendix A.2**, we follow the implementation and preset paramaters of the Dislib paper [L1] to calculate the Mutual Information (MI).  Firstly we sample a large batch (size=10000) of images and their factor labels. Then we feed these images to the models and receive 10000 samples of $(\boldsymbol{c}, \boldsymbol{v})$ pairs, where $\boldsymbol{c}$ is the representation vector, and $\boldsymbol{v}$ is the ground truth factors. To calculate $I(\boldsymbol{c}_i, \boldsymbol{v}_j)$ in Eq. 1, we first discretize  $\boldsymbol{c}_i$  by 20 bins. Since the factor $\boldsymbol{v}_j$ is naturally discrete, we use the MI formula for discrete random variables as in https://scikit-learn.org/stable/modules/generated/sklearn.metrics.mutual_info_score.html .
>
> **Q7. How to select k in partial disentanglement estimation.**
>
> **A7:** The parameter $k$ is an evaluation choice for the top-k MED. In the main text, we choose $k$ to match the dimension of VAE methods in Dislib[L1].  To show the influence of $k$, we have proposed an ablation study on $k$ in **Appendix F.3**. The conclusion is that the ranks of top-k MED are consistent with a wide range of k (see **Figure 14**).
>
> **References:**
>   - [E1] Eastwood, Cian, et al. "A Framework for the Quantitative
> Evaluation of Disentangled Representation".
>   - [L1] Locatello, Francesco, et al. "Challenging common assumptions in the unsupervised learning of disentangled representations."
>   - [K1] Kornblith, Simon, et al. "Similarity of neural network representations revisited."
>   - [H1] He, Kaiming, et al. "Momentum contrast for unsupervised visual representation learning."
>   - [W1] T. Wang and P. Isola. "Understanding contrastive representation learning through alignment and uniformity on the hypersphere"
>   - [Z1] R. S. Zimmermann, Y. Sharma, S. Schneider, M. Bethge, and W. Brendel. "Contrastive learning inverts the data generating process"
>   - [C1] Chen, Ting and Kornblith, Simon and Norouzi, Mohammad and Hinton, Geoffrey, "A Simple Framework for Contrastive Learning of Visual Representations"
>   - [G1] Jean-Bastien Grill et al, "Bootstrap Your Own Latent: A New Approach to Self-Supervised Learning"
>   - [C2] Xinlei Chen and Kaiming He, "Exploring Simple Siamese Representation Learning"
>   - [T1] Tian, Yuandong, Xinlei Chen, and Surya Ganguli. "Understanding self-supervised learning dynamics without contrastive pairs." International Conference on Machine Learning. PMLR, 2021.

---

> ### Comment · Reviewer_beE6 · 2022-08-07
> **Raise score from 4 to 5.**
>
> I appreciate the authors' responses, particularly in Table 7, Table 8, Figure 17, and Section I.2. Now, I am convinced that MED is a good metric for disentanglement. Although Section I.2. is simple, I can get an insight into the drawback of other metrics, e.g., DCI having a curse of dimensionality. Thus, I raise my score to borderline accept. If the paper was accepted, I would like to see a more complete Table 7, Table 8, and Section I.2 in the main body.

---

> > ### Author Response · Authors · 2022-08-08
> > **Thanks for the feedback**
> >
> > Thanks for recognizing the value of MED and raising the rating. We will make more detailed, complete, and formal analysis towards comparing MED and other metrics, both empirically and theoretically in the next version.

---

### Official Review · Reviewer_mxAw · 2022-07-09

**Rating:** 6
**Confidence:** 4
**Soundness:** 3 good
**Presentation:** 3 good
**Contribution:** 3 good

**Summary:**

This paper provides experiments on disentanglement for negative-free contrastive learning. The results indicate that current metrics have limitations on this setup. As a result, the authors present a novel metric for evaluating disentanglement in the proposed scenario.


**Questions:**

In section 4 the authors show how the MED metric finds sub-sample of disentangled factors, however a correspondent analysis for traditional metrics could also be performed for comparison?

**Limitations:**

The work validates the proposed metric empirically, however would be also interesting to investigate the theoretical justification of this method.

**Strengths And Weaknesses:**

Strengths.
- Investigating and expanding current metrics for quantifying disentanglement can address an important issue in deep learning.
- The assessment of current issues on disentangled metrics is done properly
- They assess disentanglement on real-datasets rather than synthetic
- Validation of the new metric with qualitative results on generative factors

Weaknesses.
Missing comparison with other recently proposed metrics.
- https://openreview.net/pdf?id=HJgK0h4Ywr
- https://openreview.net/pdf?id=EbIDjBynYJ8
- https://arxiv.org/abs/2106.03375

---

> ### Author Response · Authors · 2022-08-02
> **Response to Reviewer mxAw**
>
> Dear reviewer,
>
> Thank you so much for your review and suggestions. Below are our responses to your questions.
>
> **Q1. Missing comparison with other metrics from related works.**
>
> **A1:** Thanks for the references! For the three papers you mention:
> 1. The first paper (https://openreview.net/pdf?id=HJgK0h4Yw) proposes 4 new metrics, namely, WSEPIN, WINDIN, RMIG, and JEMMIG. Unfortunately,  for contrastive learning (CL) methods, we can not implement these metrics since the output of the CL encoders is a point estimate. They do not output the distribution $q(z_i | x)$ as in VAEs. Here we denote the output of CL methods as $o$ with other notations same as the original paper. In our paper, to calculate MIG for CL methods, VAE methods estimate posterior mean $\mathbb{E} = \mathbb{E}_{q(z_i | x)}[z_i]$ but we only have $o_i$ in CL methods. An adoption of those metrics to CL methods won't provide new information because:
>      1. For WSEPIN and WINDIN, since we can not assume the independence between $o_i$ for CL methods,  Eq. (9) in this paper does not follow. Moreover,  even given that $o_i = \mathbb{E}$, it is still impossible to estimate $I(z_i, z_{\neq i})$ by quantization since $z_{\neq i}$ is a too high-dimensional vector.
>       2. For RMIG and JEMMIG,  their key contribution is the probabilistic  assumption. However, CL methods do not take probabilistic assumptions for $o_i$. So the decomposition of $p(z_i, y_k, x^{(n)})$ is not meaningful for representations learned by CLs.   If we ignore their probabilistic background and replace $\mathbb{E}_{q(z_i | x)}[z_i]$ with $o_i$, RMIG degrades to MIG.
> 2. We noticed the SlowVAE paper (https://openreview.net/pdf?id=EbIDjBynYJ8) and included it in our comparison (**Figure 4** and more results in **Table 4 in the appendix**). They use metrics in DisLib protocol and MCC from the ICA evaluation. As we explained in the second part of Related Works, MCC evaluates identifiability, which is different from "disentanglement". On the other hand, Modularity used in SlowVAE is proven not aligned with other metrics in DisLib, so we don't include these two metrics in our evaluation. Moreover, on the method side, we can hardly include SlowVAE in all experiments because it requires temporal information to work. This is why this paper does not include standard image disentanglement datasets but uses video-based datasets such as KITTI-Masks. And to implement SlowVAE on standard image datasets, the SlowVAE paper samples the input pairs to satisfy their assumption on temporal factor transition. This sampling process requires access to ground truth factors, thus it is supervised instead of unsupervised as VAE/GAN/CL.
> 3. In the paper of  Lie Group VAE (https://arxiv.org/abs/2106.03375), four disentanglement metrics are used: FVM (FactorVAE metric), SAP, MIG, and DCI Disentanglement score. We had included these four metrics in our experiments (see **Table 4 in the appendix**).
>
> **Q2. Evaluation with traditional metrics on the subspace:**
>
> **A2:** Thanks for your suggestions. We include traditional metrics on the top-k subspaces in **Table 4**. Our observations are as follows.
> 1.  For BetaVAE and FactorVAE scores, the subspaces have better or compatible scores compared with unsupervised methods with low latent dimensions. (Note that Ada-GVAE, Ada-ML-VAE,  SlowVAE, and EBM are supervised methods.)
> 2. Regarding MIG and SAP scores, the performance of the top-k subspace is inferior. It is not surprising because MIG and SAP require a large gap between the two most relative dimensions of each factor. However, we pick k most relative dimensions out of 1000. Thus, the gap should be small.
> 3. Except for dSprites and CelebA, the top-k subspaces have lower DCI scores. The key reason behind this is the significant drop in the amount of information in the top-k process (We cut down the dimension from 1000 to 10), which drastically destroys the original encoding pattern. Consequently, the GBT trained during the DCI process tends to "borrow"  information from latent dimensions emphasizing other relevant factors to predict some complicated factors. Then the dimensions "lending" information become less disentangled. For example, to classify 183 types of cars in Cars3D from only 9 dimensions with only 3 emphasizing *object type* (note that k=3), GBT has to use information from *elevation* since these two attributes are correlated (see **Appendix B.3.**) However, for other low-dimensional methods, we keep all they have learned from training. Therefore, they are not affected by the information loss issue.
> 4. To summarize, the top-k subspaces are better than or on par with other unsupervised methods with low latent dimensions regarding FactorVAE and BetaVAE scores. Yet, for the other traditional metrics, the top-k subspaces have less impressive performance due to the mechanism of these metrics.

---

> > ### Comment · Reviewer_mxAw · 2022-08-08
> > **Thanks for the response**
> >
> > Your response clarified my question. I confirm my mark of acceptance.

---

### Official Review · Reviewer_d961 · 2022-07-10

**Rating:** 5
**Confidence:** 4
**Soundness:** 3 good
**Presentation:** 3 good
**Contribution:** 3 good

**Summary:**

This paper takes the negative-free contrastive learning methods as an example to explore the disentanglement property of the self-supervised methods experimentally. To address the limitations of existing disentangled metrics in high-dimensional representation models, the author proposes a new decoupling metric-MED based on mutual information. Experiments on real-world datasets and high-dimensional representation space demonstrate this metric's superiority and applicability.

**Questions:**

Refer to Weaknesses

**Limitations:**

The authors have not addressed the limitations and potential negative societal impact of their work

**Strengths And Weaknesses:**

Strengths:
1. Existing work on Disentangled Representation Learning is limited to the generative model. This paper empirically studies the disentanglement property of negative-free contrastive learning, which is an exploratory work.
2. This paper proposes a new metric, MED/top-K MED, which extends the decoupling metric to high-dimensional space.
3. The paper is well organized, so the reader can get the gist of the article. The authors clearly describe the proposed method. In addition, the experimental results show the effectiveness of the proposed method.

Weaknesses:
1. The author designed a version of MED/top-k MED, to evaluate the disentanglement. Section 5.4 analyzes the effect of dimension on Top-2 MED. However, the effect of dimension on MED is unclear. In the upper part of Table 1, the authors provide results in 1000-dimensional representation space. However, the paper does not directly show results in lower-dimensional spaces (no additional PCA required, just set the projection dimension), such as 100 or 200.
2. The author found that contrastive learning without negatives learned a well-disentangled subspace of latent representation from experiments. It is not yet known how this subspace and the subspace learned by disentangled representation learning perform on downstream tasks, such as classification tasks.

---

> ### Author Response · Authors · 2022-08-02
> **Response to Reviewer d961**
>
> Dear reviewer,
>
> Thank you so much for your review and suggestions. Below are our responses to your questions.
>
> **Q1: the paper does not directly show results in lower-dimensional spaces (no additional PCA required, just set the projection dimension), such as 100 or 200**
>
> **A1:** The results of MED scores for methods with variant dimensions are shown in the following tables. In the revised version, we plot the MED scores in **Appendix C.2** for methods with variant dimensions. Please see **Figure 12** for the details. The MED scores of BYOL increase as the representation dimension increases due to the inferior performance of contrastive methods with low latent dimensions. In contrast, VAEs' failure to scale to higher latent dimensions leads to a negative correlation between MED scores and latent dimensions. The conclusion for the effect of dimension on MED score is similar to that of top-k MED.
>
> | Metrics   | Models    | 8-d     | 16-d   | 128-d  | 512-d  | 1024-d  |
> |-----------|-----------|------------|-----------|-----------|-----------|-----------|
> | MED       | BYOL      | 5.7 (2.3)   | 15.0 (4.8) | 26.9 (1.3) | 31.8 (0.8) | 31.3 (0.4) |
> | Top-k MED |    BYOL       | 5.7 (2.3)   | 15.0 (5.9) | 47.8 (2.4) | 52.8 (0.9) | 53.7 (0.7) |
>
> | Metrics   | Models    | 10-d        | 100-d     | 500-d     | 1000-d    |
> |-----------|-----------|-------------|-----------|-----------|-----------|
> | MED       | BetaVAE   | 32.6 (10.0) | 3.7 (0.3)  | 2.8 (0.3)  | 2.0 (0.2)  |
> | MED       | FactorVAE | 32.5 (10.1) | 1.3 (1.2)  | 1.4 (1.8)  | 0.3 (0.3)  |
> | MED       | DIP-VAE-I | 18.8 (5.6)   | 1.2 (0.5)  | 1.4 (1.2)  | 0.1 (0.1)  |
> | top-k MED | BetaVAE   | 32.6 (10.0)  | 11.7 (1.6) | 12.6 (4.7) | 16.6 (6.2) |
> | top-k MED | FactorVAE | 32.5 (10.1)  | 2.4 (1.5)  | 4.7 (0.2)  | 3.0 (4.0)  |
> | top-k MED | DIP-VAE-I | 18.8 (5.6)   | 6.2 (2.5)  | 5.6 (3.3)  | 7.0 (1.3)  |
>
> **Q2: It is not yet known how the subspace performs on downstream tasks, such as classification tasks.**
>
> **A2:** In the revised version, we report the accuracy of predicting factor values from the whole and top-k representations by Logistic Regression.  Please refer to **Appendix F.3** and **Table 6**. We find that on 3 out of 5 datasets (dSprites, Shapes3D, and CelebA), the subspaces of contrastive learning methods have better or comparable performance compared with VAEs' full spaces. The factor predictor trained upon full space is usually better than that upon subspace, showing the advantage of high-dimensional representation again. An exception is CelebA where their performance is close, suggesting complex real-world dataset can be more robust to overfitting.

---

> ### Author Response · Authors · 2022-08-09
> **Do our responses answer your questions?**
>
> Dear Reviewer d961,
>
> It is close to the end of the author-reviewer discussion period. We would like to ask if our responses have addressed your concerns and answered your questions adequately. If not, we are happy for any further discussions.
>
> Regards,
>
> The authors.

---

### Author Response · Authors · 2022-08-02
**General Responses to Reviewers**

We sincerely thank the suggestions and comments from all reviewers. Here we emphasize the motivation, contribution, and limitations of this work to clarify some related confusion. In the original submission, due to the page limit, we included many details and extended discussion in the appendix (supplementary).

**Motivation and contribution of this work**: Current self-supervised learning is mainly advanced by empirical observations. And we noticed two topics remain unexplored: the disentangled property of negative-free contrastive learning and representation disentanglement evaluation in a high-dimensional space. These two are related because contrastive self-supervised learning requires a high dimension of representation for training. Therefore, we analyzed the existing disentanglement metrics and found their limitations when evaluating methods of different dimensions. We thus propose MED as a new metric for **fair and meaningful** disentanglement evaluation for models of different dimensions. With the tool, we can evaluate the disentanglement of negative-free contrastive learning methods for the first time. And we can benchmark contrastive learning (not just negative-free contrastive learning) with popular disentangled representation learning methods together for the first time.

**Limitations of this work**: We found two huge obstacles to measuring the disentanglement property of contrastive learning. First, there is no widely accepted definition of "disentanglement" yet, so we have to face diverse existing metrics with different designs to define and measure "disentanglement". Second, the area of contrastive learning is still advanced by empirical observations without fundamental theoretical understanding. For example, the reason why self-supervised learning can learn meaningful representation with nonlinear neural networks instead collapse even without negative samples remains a mystery. Limited by these practical constraints, this work aims to extend the disentangled representation learning study to high-dimensional space and get more understanding of representation from CL. But we can only mainly make the discussion and provide the evidence in an empirical fashion. We believe the empirical study is useful to both communities. And as said at the end of the paper draft, we hope our work can reveal some clues to motivate future theoretical justifications.

We submitted a new version of the draft, including the appendix. To clearly show the modification of this version and fit the page limit, we added the most revised content in the appendix and wrote it in **blue**.

*Changelog of the revised draft and supplementary*
1. In **Section 1** and **Section 2**, we add more illustrations of the motivations and contributions of this work.
2. We improve the writing of **Section 3.2**  to make the high-level intuition of MED clearer.
3. We provide more qualitative results:
   1. The importance matrix heatmap **(Figure  7(c))** and co-occurrence **(Figure 8(d))** matrix of dSprites including all the factors.
   2. We correct the typos in **Figure 8(a) and (c)**.
4. We add more results for quantitative evaluation.
   1. We include the existing disentanglement metrics scores on the top-k subspace in **Table 4**, with analysis in **Appendix C.1**.
   2. We plot the MED scores for different representation dimensions in **Figure 12**, covering the results for BYOL with lower latent dimensions and VAEs with higher latent dimensions.
5. We add more details for MED and top-k MED in **Appendix F**.
   1. In **Appendix F.1** we provide more explanation of the intuition and rationale of MED.
   2. In **Appendix F.2**, we discuss the limitations of previous metrics in detail and show how MED can resolve them.
   3. In **Appendix F.3**, we study more properties of top-k MED, covering the visualization of the top-k process **(Figure 13)**, ablation study on k **(Figure 14)**, and downstream factor prediction performance  **(Table 6)**.
6. We add more discussion about the disagreement of existing metrics in **Appendix G**. We redraw Figure 4 as **Figure 15** as suggested by Reviewer s3YA. But putting multiple metrics of different scales with a shared y-axis can lead to confusion. So to better explain the disagreement among existing metrics, we visualize the ranking of  methods on SmallNORB with different metrics in **Figure 16**.
7. We include the source code of MED implementation in the **supplementary**. We removed sensitive information in the submitted source code to keep anonymity.
8. We add visualization of traversing latent code in **Appendix H** on Shapes3D. The results show that MED score can be well aligned with the human sense and high-level expectation to "disentanglement".
9. We add more analysis to show the superiority of MED compared to other metrics in **Appendix I**. It includes both more comprehensive experimental support and theoretical analysis in constructed linear scenarios.

---

### Meta-Review · Area_Chair_ataN · 2022-08-21

**Recommendation:** Accept
**Confidence:** Certain

**Metareview:**

There was a consensus among reviewers that this paper should be accepted. The key convincing arguments that this paper studies a novel setting: how to measure the disentanglement in high-dimensional spaces. For this, the authors perform extensive experiments and come up with a novel metric. The reviewers further felt that concerns raised in the initial reviews were subsequently addressed in the author rebuttal.

**Award:**

No

---

### Decision · Program_Chairs · 2022-09-14

Accept